# Combining rapid antigen testing and syndromic surveillance improves community-based COVID-19 detection in a low-income country

Fergus J. Chadwick [1,2✉], Jessica Clark [1,2], Shayan Chowdhury [3], Tasnuva Chowdhury [1], David J. Pascall [4], Yacob Haddou[1,2], Joanna Andrecka[5], Mikolaj Kundegorski [2,6], Craig Wilkie [2,6], Eric Brum [5], Tahmina Shirin[7], A. S. M. Alamgir[7], Mahbubur Rahman [7], Ahmed Nawsher Alam [7], Farzana Khan[7], Ben Swallow [2,6], Frances S. Mair[8], Janine Illian [2,6], Caroline L. Trotter [9], Davina L. Hill [1,2], Dirk Husmeier [6], Jason Matthiopoulos [1,2], Katie Hampson [1,2] & Ayesha Sania[10]

Diagnostics for COVID-19 detection are limited in many settings. Syndromic surveillance is often the only means to identify cases but lacks specificity. Rapid antigen testing is inexpensive and easy-to-deploy but can lack sensitivity. We examine how combining these approaches can improve surveillance for guiding interventions in low-income communities in Dhaka, Bangladesh. Rapid-antigen-testing with PCR validation was performed on 1172 symptomatically-identified individuals in their homes. Statistical models were fitted to predict PCR-status using rapid-antigen-test results, syndromic data, and their combination. Under contrasting epidemiological scenarios, the models' predictive and classification performance was evaluated. Models combining rapid-antigen-testing and syndromic data yielded equal-to-better performance to rapid-antigen-test-only models across all scenarios with their best performance in the epidemic growth scenario. These results show that drawing on complementary strengths across rapid diagnostics, improves COVID-19 detection, and reduces false-positive and -negative diagnoses to match local requirements; improvements achievable without additional expense, or changes for patients or practitioners.

[1] Institute of Biodiversity, Animal Health and Comparative Medicine, University of Glasgow, Glasgow, UK. [2] COVID-19 in LMICs Research Group, University of Glasgow, Glasgow, UK. [3] a2i, United Nations Development Program, ICT Ministry, Dhaka, Bangladesh. [4] MRC Biostatistics Unit, University of Cambridge, Cambridge, UK. [5] Food and Agriculture Organisation of the United Nations in support of the UN Interagency Support Team, Dhaka, Bangladesh. [6] School of Mathematics and Statistics, University of Glasgow, Glasgow, UK. [7] Institute of Epidemiology, Disease Control and Research, Ministry of Health, Dhaka, Bangladesh. [8] General Practice and Primary Care, Institute of Health and Wellbeing, University of Glasgow, Glasgow, UK. [9] Departments of Pathology and Veterinary Medicine, University of Cambridge, Cambridge, UK. [10] Division of Developmental Neuroscience, Department of Psychiatry, Columbia University, New York, NY, USA. ✉email: fergusjchadwick@gmail.com

dentification and isolation of COVID-19 cases remains key to the pandemic response. The faster and more accurately cases can be identified, the more effectively clinical care can be provided, and transmission reduced through targeted interventions. Real-time PCR has rapidly become the gold-standard test for SARS-CoV-2 detection (although Dramé et al. point out that, with less than 100% sensitivity, PCR falls short of being a true gold-standard)[1] due to its high sensitivity and specificity[2]. However, turnaround can be slow and access to laboratory diagnostics is limited in many parts of the world. As such, syndromic surveillance has often been the primary means of case identification for guiding individual and population-wide mitigation measures[3,4]. Rapid antigen tests are an increasingly popular alternative to PCR as they have high specificity, and are less expensive, easier to perform, and faster, returning results within 20 min. Hence, rapid antigen tests have potential to greatly decrease the time and expense associated with case detection, but concerns have been raised that their lower sensitivity leads to unacceptably high false-negative diagnoses[5–8]. Improving COVID-19 diagnosis is a priority and, therefore, requires us to better harness imperfect but fast and inexpensive methods, particularly for individual diagnosis but also for population-level surveillance[9].

Syndromic surveillance has been used since the start of the pandemic[10]. The COVID-19 case definition was based on early data from clinical cases[11], but, as the virus has evolved and spread, the clinical picture of COVID-19 has changed. Updated case definitions have improved, though are necessarily non-specific and generate many false-positive diagnoses (and ignores asymptomatic cases entirely)[12,13]. A natural extension is syndromic modelling, whereby symptomatic and risk factor data are used to fit a model to allow more accurate prediction of how likely a patient is to have COVID-19[14]. However, disease syndromes change between populations, when new variants emerge, and as other diseases become more or less common[12,15], which can make syndromic models perform poorly in new settings across space and time. This is a particular challenge for seasonal respiratory pathogens, where symptoms often co-occur and are non-specific[12].

A key limitation of both rapid tests and syndromic surveillance is their low effectiveness at COVID-19 detection in asymptomatic patients. Asymptomatic cases are known to play a role in driving transmission[16]. Resource limitations mean that many health agencies and governments have exclusively or temporarily targeted interventions towards symptomatic individuals to reduce transmission. Asymptomatic cases can still be identified through contact tracing from symptomatic patients. Reliable diagnosis of symptomatic cases of COVID-19, therefore, is a priority in many settings and is the focus of this paper.

Even for symptomatic patients, neither rapid tests nor syndromic surveillance can match PCR in terms of both sensitivity and specificity. However, lower sensitivity and specificity may be admissible depending on the scale and impact of misclassification[17]. Indeed, there are costs to both individuals and societies that must be considered when making policy decisions to determine the most appropriate approach to testing. Low specificity means more common COVID-19 misdiagnoses (false positives), leading to unnecessary self-isolation, which is expensive to individuals and society[18]. Low sensitivity means COVID-19 cases will be missed (false negatives) and mitigation measures not put in place leading to increased transmission and disease burden[19]. These misclassifications are complementary for a given diagnostic, meaning increasing specificity will lead to decreased sensitivity, and vice versa.

The typical approach is to balance sensitivity and specificity to maximise the number of correct classifications and assume that both misclassification types are equally costly. The costs of false positives and false negatives, however, vary enormously depending on the intersection of perspective, economic and epidemiological concerns. An individual may be motivated to secure a false-negative diagnosis if there is insufficient support for self-isolation. In contrast, at the government level, false positives may be acceptable if the economic cost of supporting those individuals is less than the cost of accelerating case rates. The epidemiological context will also alter the impact of false positives and false negatives. For example, if the disease is prevalent or increasing the priority of both individuals and governments may be to curb transmission and reduce impacts as quickly as possible. In this instance, false negatives have an outsized and costly impact by increasing the number of contact events occurring in the population and delaying control measures by underestimating epidemic size[19]. In contrast, under low prevalence, false negatives will be correspondingly low so even a high false-negative rate (low sensitivity) will have modest impact, but small decreases in specificity will lead to a large number of expensive false positives[20]. In practice, the situation will be more nuanced and modulated by testing capacity constraints, requiring a balance to be struck[17].

The best diagnostic approach for surveillance will therefore be one where correct classifications have highest value and misclassifications have lowest cost. Here, we examine the use of rapid antigen testing and syndromic surveillance of COVID-19 in symptomatic patients from low-income communities in Dhaka, Bangladesh, where a large volunteer workforce supports COVID-19 diagnosis, care and prevention. In this context, community-based workers used a mobile-phone-based application to record patient symptoms and provide advice and support services, with a diagnostic algorithm deployed on the app to inform their provisioning. This algorithm could be updated in real time depending upon the epidemiological context to allow appropriate tailoring of service provision (although was not updated during the study period).

Here, we demonstrate that by combining rapid antigen testing and syndromic surveillance we can draw on their complementary strengths, ameliorate their respective weaknesses and tune them for different epidemiological scenarios. We compare their performance alone and in combination for general prediction and as diagnostics under three scenarios with different misclassification requirements determined by government policy-makers. Overall, we show that the optimised combined models achieve equal-to-much-lower error rates than the rapid antigen test- or syndromic surveillance-only in all metrics, and how integrating data from multiple rapid testing methods can improve diagnostics, particularly when adapted to local situations.

## Results

**Population characteristics**. Of 1241 participants enrolled by community support teams (CSTs) across Dhaka, 1172 (94%) had complete data available for analyses. The remainder were removed due to duplicated sample identification codes that prevented reliable matching of test results to symptom metadata. These duplications occur at random, due to human error, and we do not believe they could bias results. Patient summaries by age, gender, case positivity and symptoms are presented in Table 1. No participants had been vaccinated as the study pre-dated mass vaccination in low-income communities in Dhaka and only symptomatic patients were included in this study because they were the local government priority for support. Case positivity measured by PCR in Dhaka increased from 15.8% to 23.8% from the first (19th–26th May 2021) to the last week (4th–11th July 2021) of the study, corresponding to prevalence rising from 1.4 to 13.8 confirmed cases per 100,000 people[21].

**Table 1 Breakdown of patient numbers by age and gender, in relation to case positivity by PCR and reported symptoms (both as % rounded to nearest integer).**

| Age (years) | Gender | Count | Positivity rate (%) | Symptoms (%) | | | | | | | | | | | | | | |
| --- | --- | --- | --- | --- | --- | --- | --- | --- | --- | --- | --- | --- | --- | --- | --- | --- | --- | --- |
| | | | | Breathing problems | Cough (any) | Cough (dry) | Cough (wet) | Diarrhoea | Ongoing fever | Headache | Loss of smell | Loss of taste | Muscle pain | Red eyes | Runny nose | Sore throat | Tiredness | Vomiting |
| 16-25 | Women | 124 | 19 | 23 | 73 | 69 | 19 | 4 | 94 | 77 | 38 | 51 | 52 | 10 | 49 | 43 | 73 | 19 |
| 16-25 | Men | 157 | 20 | 20 | 74 | 72 | 22 | 5 | 91 | 73 | 44 | 45 | 50 | 10 | 36 | 42 | 62 | 13 |
| 26-35 | Women | 144 | 17 | 25 | 72 | 70 | 19 | 10 | 90 | 75 | 35 | 42 | 51 | 4 | 40 | 43 | 69 | 7 |
| 26-35 | Men | 178 | 26 | 26 | 80 | 78 | 14 | 10 | 89 | 74 | 38 | 38 | 49 | 7 | 38 | 33 | 69 | 16 |
| 36-45 | Women | 101 | 26 | 28 | 79 | 77 | 25 | 4 | 93 | 78 | 38 | 48 | 53 | 5 | 47 | 42 | 72 | 18 |
| 36-45 | Men | 119 | 24 | 23 | 75 | 71 | 18 | 7 | 89 | 71 | 38 | 38 | 55 | 8 | 39 | 41 | 67 | 8 |
| 46-55 | Women | 66 | 20 | 17 | 74 | 74 | 15 | 3 | 86 | 70 | 32 | 32 | 55 | 0 | 35 | 33 | 58 | 15 |
| 46-55 | Men | 58 | 22 | 16 | 55 | 55 | 14 | 2 | 84 | 57 | 34 | 34 | 52 | 10 | 45 | 33 | 69 | 7 |
| 56+ | Women | 57 | 23 | 25 | 72 | 68 | 23 | 11 | 84 | 54 | 33 | 30 | 49 | 4 | 32 | 26 | 60 | 14 |
| 56+ | Men | 61 | 26 | 30 | 66 | 64 | 15 | 5 | 77 | 59 | 41 | 36 | 49 | 8 | 36 | 23 | 52 | 11 |
| All | | 1065 | 22 | 23 | 74 | 71 | 19 | 7 | 89 | 71 | 38 | 41 | 51 | 7 | 40 | 38 | 66 | 13 |

Although age is binned here, raw age in years was used for analyses. Furthermore, in the survey non-binary genders were permitted but none reported.

**Model selection**. Backwards model selection using strength of posterior correlation with outcome (Methods: Statistical modelling: model selection) for both the multivariate probit syndromic data only model and the thresholded multivariate probit syndromic data with rapid antigen test result (hereafter the Syndromic-only and Syndromic-RAT combined) models showed a marked decline in predictive power at more than four symptoms. The final four symptoms retained in Syndromic-only were loss of smell, ongoing fever, diarrhoea and loss of taste and in Syndromic-RAT combined were ongoing fever, wet cough, loss of smell and dry cough. The symptoms are listed in reverse order of importance as determined by model selection (i.e., all four symptoms were retained in the four symptom model, the first was removed in the three symptom model, the second was also removed in the two symptom model) and the median estimated correlations can be seen in the Supplementary Results (Supplementary Figures 1 and 2). The covariate gender was dropped for both model classes while age was dropped in the Syndromic-RAT combined class but retained in the Syndromic-only class.

**Predictive performance**. In the comparison of predictive performance under out-of-sample temporal cross-validation ('Methods: Statistical modelling: model performance'), RAT-only (rapid antigen test result) performed worst with a cross-entropy of 3.18 (cross-entropy values further from zero correspond to worse predictive performance). The median cross-entropy values were between 2.71 and 2.78 for Syndromic-only models. Syndromic-RAT combined models performed best with cross-entropy values between 1.56 and 1.6 (Fig. 1).

**Classification performance**. Generic model classification performance under out-of-sample temporal cross-validation ('Methods: Statistical modelling: model performance') for the one and four symptom models in the Syndromic-only and Syndromic-RAT Combined classes is shown by their ROC curves (Fig. 2). The curves for the models of different complexities are extremely similar (as are the two and three symptom model curves, not shown), however, note that the four symptom model has higher precision and granularity across both axes. The RAT-only model is a binary test (rapid antigen test positive or negative) and so the ROC is a single value, not a curve, with a false-positive rate of 0.02 and a false-negative rate of 0.45.

**Scenario-specific performance**. Scenario-specific classification performance under out-of-sample temporal cross-validation ('Methods: Statistical modelling: model performance') is shown in Fig. 3. Across all scenarios (defined in Table 2), the best models in Syndromic-RAT Combined that used both the rapid antigen testing and syndromic data performed equally well or better than the other two model classes. In Scenario 1 ('Agnostic', wherein the correct classification is maximised, assuming equal costing of false positives and false negatives, Table 2), models in RAT-only and Syndromic-RAT Combined classes performed equally well (overlapping posterior interquartile ranges) and distinctly better (no overlap in posterior interquartile range) than models in the Syndromic-only class. The median errors, as defined in Table 2, were 0.43 for models in RAT-only and Syndromic-RAT Combined and between 0.85 and 0.86 for Syndromic-only models (Fig. 3). In Scenario 2 ('Epidemic Growth', wherein false-negative rates must be below 20%, Table 2), the RAT-only models failed to meet the scenario requirement. The median errors were between 0.74 and 0.75 for Syndromic-only models and 0.41 and 0.5 for Syndromic-RAT Combined models (Fig. 3).

In Scenario 3 ('Declining Incidence', wherein false-positive rates must be below 20%, Table 2), Syndromic-only again performed

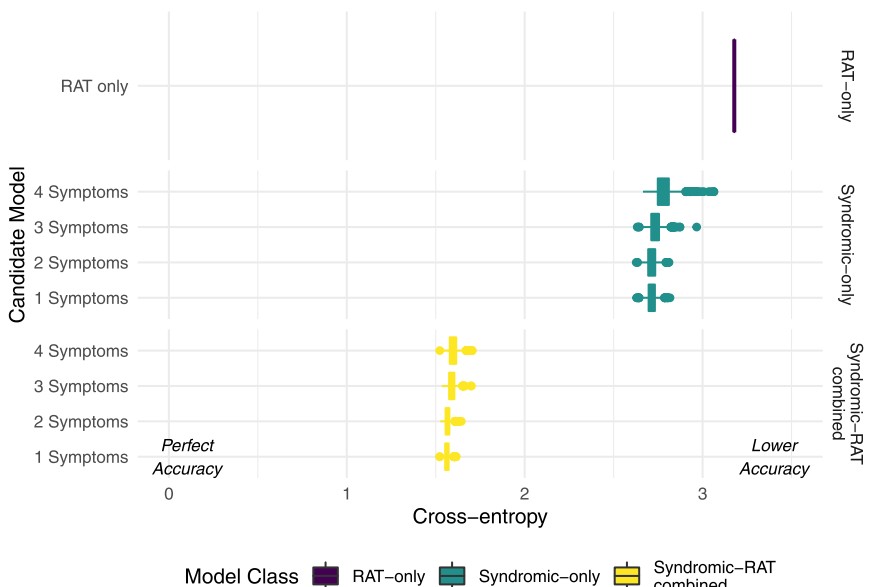

**Fig. 1 Model predictive performance.** Predictive performance of candidate models was measured using out-of-sample cross-entropy. Combined posterior median and interquartile ranges for $n = 1172$ biologically independent individuals predicted under temporally structured cross-validation. Cross-entropy shows the most generalised-level of model predictive power, assessing performance in the probability scale without requiring classification threshold decisions. A cross-entropy of zero indicates a model that predicts with certainty the correct result each time. A random classifier for the problem scored 11.54. Interquartile ranges are shown for the posterior cross-entropy of the best candidate models at each level of model complexity tested under temporal cross-validation. The intermediate complexity models perform best at prediction, although performance is similar across all the models within each model class. There was a marked decline in predictive power at more than four symptoms, leading us to choose this as the maximum complexity model in our candidate models. Model classes are colour-coded, the rapid antigen test only (RAT-only) model is purple, Syndromic-only model is teal, and the Syndromc-RAT Combined model is yellow.

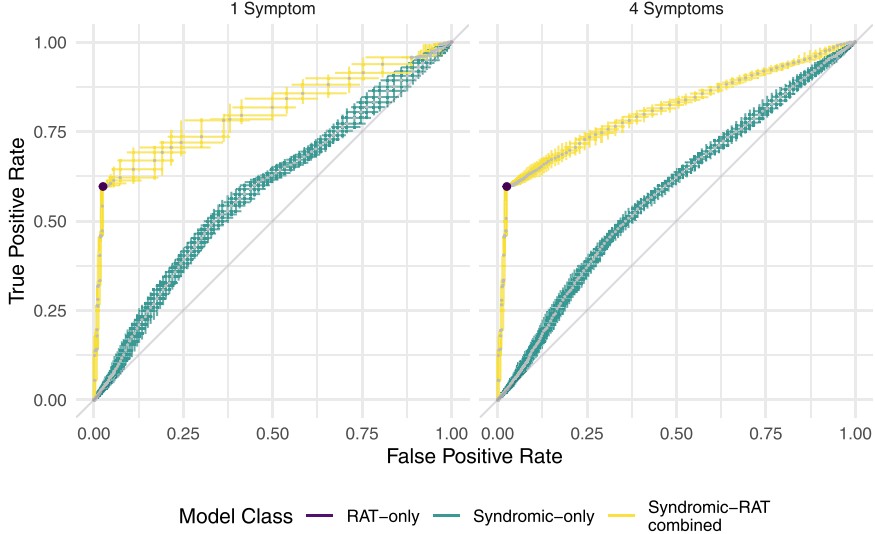

**Fig. 2 Generic model classification performance.** Median (grey dots) and interquartile ranges for receiver operating characteristics (ROC) for rapid antigen testing only approach (purple) and posterior median and interquartile range ROC for Syndromic-only (teal) and Syndromic-Rapid Antigen Test (RAT) Combined (yellow) models for $n = 1172$ biologically independent individuals predicted under temporally structured cross-validation. In the RAT-only model, the ROC is a single value (i.e., a dot rather than a line) as the binary test has a single sensitivity and specificity. In Syndromic-only and Syndromic-RAT Combined classes, the ROC values demonstrate the performance of the model for any hypothetical scenario as defined by the axes (as opposed to Fig. 5 which demonstrates model performance in specific epidemiological scenarios which are realisations of single points in this space). While ROC plots are often plotted as curves, we do not have continuous probability values due to the binary nature of predictor symptoms. This is important as discontinuity in the probabilities impacts the sensitivity of the model to classification thresholds, such as those used in the scenarios below.

worst, and Syndromic-RAT Combined achieved the lowest error, with RAT-only falling between the two (closer to Syndromic-RAT Combined than Syndromic-only). The error in RAT-only was 0.03 and the median errors ranged from 0.19 to 0.2 for Syndromic-only, and 0.19 to 0.2 for Syndromic-RAT Combined (Fig. 3). The results

for each scenario-model combination can be translated into numbers of misclassifications per 1000 tests if the test positivity rate is known. We present this in Supplementary Results (Supplementary Results Table 1) for low (5%), average (20%) and high (35%) test positivity rates in Bangladesh.

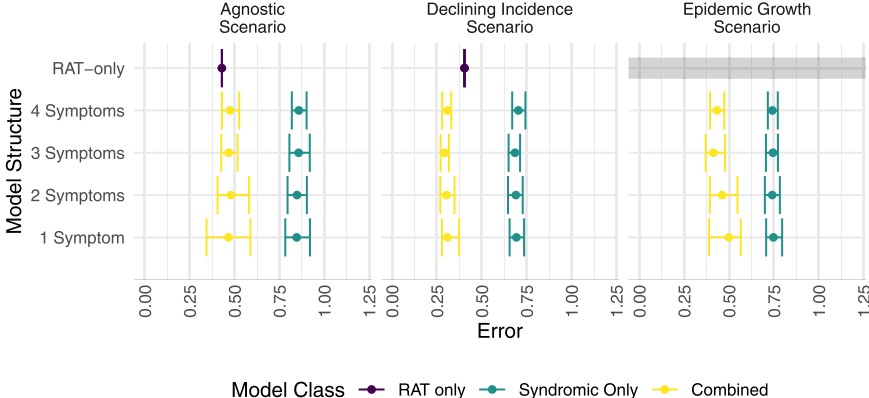

**Fig. 3 Performance of models under three epidemiological scenarios.** Combined posterior median and interquartile ranges of error rates for $n = 1172$ biologically independent individuals predicted under temporally structured cross-validation. In the Agnostic Scenario, the model is maximises the correct classification rate with error measured as the sum of the false-positive and false-negative rates. In the Epidemic Growth Scenario, a maximum false-negative rate of 20% is permitted, and the error is measured as the false-positive rate. In the Declining Incidence scenario, a maximum false-positive rate of 20% is permitted, and the error is measured as the false-negative rate. These requirements were determined through discussion with colleagues at the Institute of Epidemiology and Disease Control (IEDCR), Bangladesh. The plot shows the posterior median and interquartile range for scenario-specific errors. Lower errors correspond to better model performance. There is no error rate defined for the rapid antigen testing only model (RAT-only) in the Epidemic Growth Scenario as the model failed to meet the requirement for that scenario (indicated by grey bar). Model classes are colour-coded, the RAT-only model is purple, the Syndromic-only model is teal, and the Syndromc-RAT Combined model is yellow.

**Table 2 Requirements and performance criteria for each epidemiological scenario.**

| Scenario name | Requirement | Performance criterion (error) |
|---|---|---|
| 1 Agnostic | Maximise correct classification rates | Sum of error rates |
| 2 Epidemic growth | <20% false-negative rate | False-positive rate |
| 3 Declining incidence | <20% false-positive rate | False-negative rate |

The requirement refers to a base level of performance the model must achieve, allowing the more flexible models to be adapted to meet that requirement as closely as possible (e.g., by determining a classification threshold). These requirements were determined through discussion with colleagues at the Institute of Epidemiology and Disease Control (IEDCR), Bangladesh, using internal resource projections. The performance criterion is used to determine which model performs the 'best' given that the requirement has been met.

The candidate models are chosen as a result of a selection process and performed much better than more complex models (i.e., with 5 or more symptoms) or simpler models (with no symptoms but an intercept and age and gender as covariates) in terms of cross-entropy and ROC. For the models that used syndromic data, across all scenarios, within the final four candidate models the number of symptoms included made relatively little difference in terms of median performance (with respect to error, Fig. 3 scenario-plot and Table 2), although the more complex models have higher precision.

Across all metrics, the rapid antigen test result is the most informative data-type for potential COVID-19 patients. However, incorporation of even one symptom and the use of a modelling framework greatly improves our ability to predict and classify cases, both generically and in specified scenarios. Including additional symptoms and covariates provides further information on the patient's status and greater model flexibility, resulting in higher precision in predictions and classifications.

## Discussion

We have demonstrated that combining rapid antigen tests with syndromic modelling yields better identification of COVID-19 cases than either diagnostic in isolation. These gains in performance are mirrored across metrics of prediction, as well as general and scenario-specific classifications. The biggest improvement is seen under the scenario of 'Epidemic Growth' (Table 2), such as might be expected following relaxation of restrictions and with the emergence of new variants. In this scenario, the combined data model has a false-negative rate of 18% (IQR: 21–15), 22 (IQR: 19–25) percentage points lower than the rapid antigen test only model. Although the syndromic model matches the combined model's false-negative rate, its false-positive rate is 41% (IQR: 47–37), 33 (IQR: 30–33) percentage points higher. In real terms, at the end of our study, there was a 20% case positivity rate in Bangladesh. By applying our framework under the 'Epidemic Growth' scenario, for every 100 rapid antigen tests, our approach would capture an additional 7 cases. In a country deploying millions of tests per week, this results in catching tens of thousands of cases that would otherwise be missed. Similarly, the combined model class performs equally well or better than the other models for the other scenarios explored (Fig. 3). These scenarios offer snapshots of performance, while the model prediction and classification metrics provide an indication of how the models perform more generically (Figs. 1 and 2, respectively). The more complex model classes achieve this top performance across all scenarios and metrics measured here thanks to their flexibility (allowing them to be readily adapted to new situations) and their synergistic use of the higher specificity rapid antigen testing and the more sensitive syndromic data.

The final symptoms and covariates chosen through model selection should be interpreted cautiously. Firstly, the power of the models to detect relationships will be partially determined by sample size. Secondly, these models were developed for prediction and classification in a unique sub-population: CST-identified, symptomatic patients in low-income communities in Dhaka. From the same symptom and risk factor set, different variables were retained for different model classes, despite data being collected over a short period from the same population. These differences may point to mechanisms by which CST-identified and rapid antigen test-positive individuals differ from other groups. They also underline the importance of collecting a relatively broad range of symptom data as the syndromic profile of the disease shifts between populations. Of interest is whether individuals identified by PCR but missed by rapid antigen tests

are less infectious and more typical of asymptomatic cases (perhaps due to different lengths of time since symptom onset). This could be examined using viral load measured as threshold cycle (Ct) values from PCR and further testing for other illnesses[22]. Our use of PCR as a validation test should also be explored further, as it does not have 100% sensitivity so additional validation tests may be informative. However, finding alternative gold-standard tests that can be carried out in the community is challenging [23].

The modelling frameworks allow for the potential inclusion of additional covariates where they are collected reliably. These covariates may define different sub-populations in which we expect the relationships between symptoms and infectious status to differ. For example, vaccinated patients would be expected to exhibit fewer and milder symptoms than unvaccinated patients. By including vaccination status alongside symptoms within the model, the model can share information between the two groups while allowing the relationships to differ where this improves prediction. Similar approaches could be taken to incorporate rapid antigen test manufacturer, recent disease prevalence or time since symptom onset. Furthermore, using a modelling framework allows explicit estimation and exploration of these differences, rather than relying on post hoc analysis of misdiagnosis rates (e.g., Babu et al.[24]). When a particular data source is found to have good predictive power, it would be useful to identify whether this could target further data collection. For example, the low false-positive rate of rapid antigen tests means that, if affordable, serial testing of the same individual could increase true positive detections without a major impact on accuracy.

The boost in diagnostic performance we found was achieved by harnessing data collected by community-based health workers using a mobile-phone-based application to record patient symptoms and test results. These data were already being collected in Bangladesh and similar methods are being rolled out in other low- and middle-income countries[25,26]. We ensured our method is scalable by developing it using a large community-based sample and with input from the CST programme organisers. As CST data are collected via a mobile phone application the diagnostic model can be updated in real-time. The algorithm of the app could therefore be modified to reflect local epidemiological requirements, local case rates and the considered cost/benefits of misdiagnosis, thereby facilitating adaptation to new variants or even new diseases. Similarly, if a source of data becomes unavailable then the underlying model can be changed to reflect this. For example, if there are rapid antigen test supply problems, the app could deploy Syndromic-only which uses the same data as Syndromic-RAT Combined, without relying on the rapid antigen test, and the combined model could be retrained on tests from different manufacturers with different performance characteristics.

One of the key innovations of this framework is the ability to adapt the diagnostic to local populations and their needs. To achieve this, we need good quality, local data collection and to understand the costs of sensitivity and specificity. The costs of false negatives and false positives vary greatly depending on epidemic context, and balancing the treatment of individuals with control of the health burden at a societal level[27]. Similarly, the market price of interventions can fluctuate depending on demand, aid funding and global trends[28]. In practice, the costs of rapid antigen tests are likely to be up to an order of magnitude lower than PCR when considering the additional infrastructure and personnel. Access to testing (RAT or PCR) needs to be considered as part of weighing up the costs and benefits of surveillance approaches[29]. Understanding how to measure and balance these demands requires insights from economists, epidemiologists, social scientists and policy-makers, and is an

area of active research[30]. Given the degree of complexity, it is tempting to rely on methods that do not openly require a decision to be made about the relative costs of the different misclassification types. However, rather than removing the complex cost structures involved, such methods simply hide them. All methods place a balance on false positives and negatives implicitly, our hope is that by requiring these decisions to be made explicitly, they are more readily challenged, researched and improved upon. Similarly, the need for local data collection should not be seen as a weakness of the method, but rather a welcome requirement that allows us to directly assess intervention success and biases.

Pandemic management can only be done with testing at scale. The combined syndromic and rapid antigen testing approach that we report is promising for large-scale COVID-19 testing in low-income communities. Moreover, our framework is adaptable, including for many other infectious diseases where strict adherence to gold-standard laboratory diagnostics greatly limits testing capacity. Imperfect diagnostics are frequently imperfect in different ways, and these differences are ripe for statistical treatment. These methods are often more agile than gold-standard diagnostics in changing situations as experienced during the pandemic, when fast responses are essential. Overall, our approach shows that by understanding how to utilise the complementary strengths of imperfect but rapid diagnostics (and deploying the more limited gold-standard testing for validation), good quality large-scale testing can be achieved even in low-income communities.

## Methods

**Data collection**. Recruitment took place across low-income communities in Dhaka North Community Corporation between 19 May 2021 and 11 July 2021. Participants were identified for COVID-19 testing by CSTs. CSTs are community-based volunteer health workers trained to identify individuals reporting symptoms suggestive of COVID-19 through hotline calls or community-based reporting channels. Probable cases identified by CSTs are counselled to isolate for 14 days under household quarantine, connected to telemedicine services for home-based COVID-19 management, and provided with over-the-counter medication or medical referrals if the case is severe. CSTs submit surveillance data to a centralised database through a mobile-phone-based application (Supplementary Materials (Data Collection)).

Participants were selected for testing if they were over 15 years old, had a fever (>38 °C) at the point of assessment, and one or more of 14 symptoms listed in Table 1. CSTs collected the enrolled individual's age and gender, and took two nasal swabs. One swab was used for rapid antigen testing (SD Biosensor STANDARD™ Q COVID-19 Ag Test BioNote) at the household, and the other returned under cold-storage to the Institute of Epidemiology, Disease Control and Research (IEDCR) for PCR testing. The full questionnaire and testing protocols are provided in Supplementary Methods.

Participants provided written informed consent to sample collection and for their results to be analysed in the study. The study protocol was approved by the Institutional Review Board at the IEDCR, Ministry of Health, Bangladesh, IEDCR/IRB/04.

## Statistical modelling

*Structure*. We developed three model classes using: (1) the rapid antigen test result; (2) the syndromic data, and (3) the two data sources combined (Fig. 4). We identified cases by PCR. As RAT-only used the rapid antigen test result, no statistical model is needed. For Syndromic-only, we used a Bayesian multivariate probit model[31], with multivariate referring to multiple response variables. The multivariate probit structure allows the model to account for the binary and correlated nature of the symptoms, while conditioning on the risk factors of age and gender, thereby improving over models which implicitly assume independence between symptoms. By using a Bayesian formulation, we generate full posteriors for our parameter estimates, allowing natural quantification of uncertainty. We chose minimally informative priors, with standard normals for the covariates and intercepts and a flat LKJ distribution for the correlation matrix (described in more detail in Supplementary Materials: Statistical Methodology).

For Syndromic-RAT Combined, we use a hurdled multivariate probit. The approach exploits the specificity of rapid antigen tests by treating rapid test-positives as cases. While this sounds like a strong assumption, this simply translates in practice to telling all rapid test-positive individuals to assume they have COVID-19. Rapid antigen test-negative individuals are then modelled using the sensitive

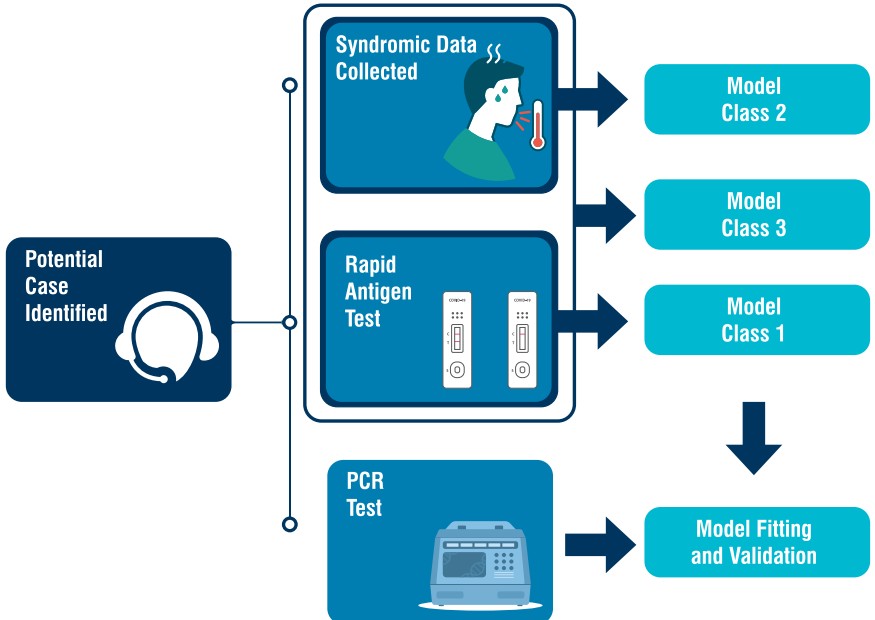

**Fig. 4 Schematic description of identification of likely COVID-19 cases by community support teams (CSTs) and model definitions.** CSTs collect syndromic data (age, gender and presence/absence of 14 predetermined symptoms), and two sets of naso-pharyngeal swabs (for rapid antigen testing and PCR). We used three model classes: rapid antigen test only in 1, syndromic data only in 2, and both rapid antigen test and syndromic data in 3. The PCR result is used to train and test each model using temporal cross-validation.

syndromic approach of Syndromic-only to capture PCR-positives missed by the rapid antigen test. This approach leverages the potentially different syndromic profiles of PCR-positive patients who are rapid antigen test positive and negative, allowing the model to adapt solely to the latter. The models were fitted to the data using Bayesian inference techniques based on Hamiltonian Monte Carlo in the Stan programming language[32]. Further technical details and model equations are presented in Supplementary Methods.

*Model selection.* For model selection and all measures of performance, we used out-of-sample, temporal cross-validation (Fig. 5), where training and testing data are separated based on time. We structured the cross-validation temporally to reflect the real-world prediction problem: using recent testing data to predict new cases. Due to the changing nature of the disease and its management over time, using unstructured cross-validation would result in an overstatement of model performance.

We conducted backwards model selection, starting with the most complex biologically plausible model, to identify a subset of models with the highest predictive power. Shrinking the number of possible models was necessary to lower computational demand and reduce the risk of overfitting. The large number of symptoms corresponds to many potential model configurations (>131,000 for 14 symptoms and 2 covariates) which might perform well on the test sets by chance (even under temporal cross-validation) but lack transferability to novel situations. The Bayesian multivariate probit structure common to these models directly estimates the full posterior correlation matrix for the PCR status and other symptoms. By first using the strength of the correlation with the PCR status (coarse selection, Fig. 5) and general predictive power (fine selection, Fig. 5) to narrow down the number of candidate models, and then testing those models under the epidemiological scenarios, we are more likely to choose models that generalise well to new data (Supplementary Materials: Statistical Methodology).

*Measuring model performance.* We assessed models using three sets of increasingly policy-relevant criteria. First, we use predictive performance to measure model performance in a decision-free context (i.e., comparing predicted probabilities of an individual having COVID-19 to their true status). Second, we use receiver operating characteristic (ROC) curves to show generic model classification performance. Finally, we measure classification performance under three epidemiological scenarios (defined in Table 2).

We scored the models' predictive power using cross-entropy (defined in Supplementary Methods). Cross-entropy measures the accuracy of predicted probabilities of binary outcomes, rather than making binary classifications, similar in concept to a mean square error for normally distributed data, but adapted for binary data[33]. A cross-entropy of zero indicates a model that predicts with certainty the correct result each time. A random classifier for the problem scored 11.54

In practice, models are often evaluated on their performance as deterministic classifiers rather than as stochastic prediction engines (i.e., their ability to classify

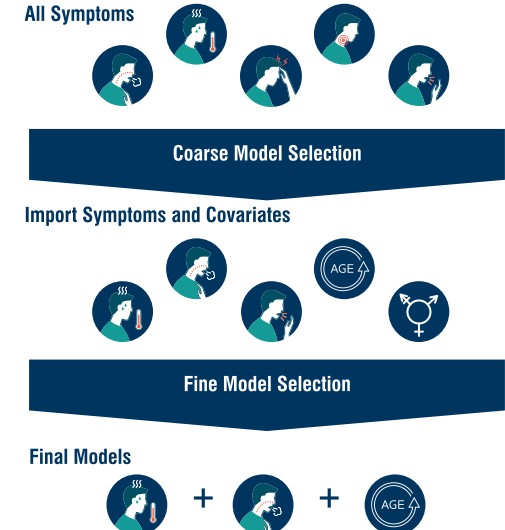

**Fig. 5 Model selection procedure.** Rounds of model selection in the multivariate probit component of the Syndromic-only and Syndromic-Rapid Antigen Test (RAT) Combined models. With 14 symptoms (5 shown for demonstration purposes) and 2 covariates there are over 131,000 possible model combinations. To make exploring these models computationally feasible and to reduce the risk of overfitting, we carried out two rounds of model selection. A subset of symptoms are identified using the strength of posterior correlation between each symptom and PCR status identified by the corresponding model, with the weakest correlated symptoms removed during each round of selection. From this subset of symptoms, a more exhaustive search of potential models is then conducted to identify the best symptom-covariate relationships, using temporal cross-validation to measure model performance. The best model for each level of complexity (i.e., number of symptoms) are then used as our candidate models. Only these final models are used for classification. This reduces the set of models tested as classifiers from >131,000 to just four per model class.

an individual as a COVID-19 case or not, rather than the probability that the individual is a case). Deterministic classification requires that a probability threshold is chosen over which patients are classified as COVID-19 positive. Classifier performance was compared generically (using ROC curves to look at the error rates that can be achieved with each model without specifying a scenario). Generic performance here is only used to show the flexibility of the model classes, i.e., model performance without reference to a specific scenario. The best model for a local situation can only be determined if the relative costs of false positives and negatives are considered.

We compare model performance under three scenarios (using error terms described in Table 2) developed for illustrative purposes through discussion with colleagues at IEDCR. In Scenario 1, we do not consider epidemiological context but minimise false-negative and false-positive rates equally by maximising the correct classification rates individually and in total, as measured by the harmonic mean (not the arithmetic mean which would maximise the rates in total, Supplementary Methods). Scenario 2 corresponds to epidemic growth as experienced during the spread of the Delta variant during the period of data collection. Under these circumstances, false negatives are costly relative to false positives. In Scenario 3, incidence is assumed to be low and relatively stable. In this situation, policy-makers may prioritise keeping false-positive diagnoses low to prevent fatigue and to keep the workforce active.

**Reporting summary**. Further information on research design is available in the Nature Research Reporting Summary linked to this article.

## Data availability

The anonymised patient data generated in this study and model outputs have been deposited in the GitHub repository fergusjchadwick/COVID19_SyndromicRAT_public[34]. The raw patient data are protected and are not available due to data privacy laws.

## Code availability

The statistical code used in this study are available in a GitHub repository at https://github.com/fergusjchadwick/COVID19_SyndromicRAT_public.

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

## Acknowledgements

This work is supported by a grant from the Bill and Melinda Gates Foundation to FAO (INV-022851). F.J.C. is funded by EPSRC (EP/R513222/1), D.J.P. by the JUNIPER consortium (MR/V038613/1) and K.H. by Wellcome (207569/Z/17/Z). We would like to thank the members of the community support teams in Bangladesh who have provided essential services throughout the pandemic. Earlier drafts of this manuscript benefited from the input of Paul Johnson, Daniel Haydon, Anne-Sophie Bonnet-Lebrun, Luca Nelli, Crinan Jarrett, Rita Claudia Cardoso Ribeiro, Halfan Ngowo, Heather McDevitt and Gina Bertolacci. The University of Glasgow COVID-19 in LMICs Group provided the environment in which to develop this work.

## Author contributions

Authors listed alphabetically. F.J.C., D.H, K.H. and J.M. conceptualised this study. J.A., F.J.C., S.C., Y.H. and M.K. curated the data. F.J.C., D.H., J.M. and D.J.P. conducted the formal analysis. E.B., K.H. and A.S. acquired the funding for the study. J.A., T.C., M.K., A.S. and F.Z. undertook the investigation. A.N.A., A.S.M.A., E.B., F.J.C., S.C., T.C., D.H., K.H., J.M., D.J.P., M.R., A.S., T.S. and C.T. developed the methodology. A.N.A., J.A., E.B., K.H., M.R., A.S., T.S. and F.Z. administered this project. A.N.A., E.B., F.J.C., S.C., T.C., F.K., M.K., J.M. and T.S. provided resources for the study. F.J.C., D.H., M.K. and D.J.P. developed the software used. A.S.M.A., A.N.A., E.B., D.H., K.H., J.I., F.K., J.M., M.R., A.S. and T.S. provided supervision. D.L.H., J.I., F.M., B.S. and C.W. validated analyses, data

and methodology used. F.J.C., J.C., C.W., D.H., D.L.H., K.H., Y.H., M.K., F.M., J.M. and D.J.P. designed the visualisations used. F.J.C. wrote the original draft of the manuscript. All authors participated in reviewing and editing the manuscript.

## Competing interests

The authors declare no competing interests.
