## [Peer Review File · Nature Communications]

Combining rapid antigen testing and syndromic surveillance improves community-based COVID-19 detection in a low-income countryREVIEWER COMMENTS

Reviewer #1 (Remarks to the Author):

Chadwick et al. used a machine learning approach to determine which symptoms best identify a COVID-19 case (equivalent to RT-PCR positive) with and without the aid of a rapid antigen test. In the discussion, the authors mention that the final symptoms and covariates are likely specific to the small unique sub-population examined, which is a drawback for the broad applicability of their particular results from the syndromic surveillance. However, the approach and methodology used can be applied to improve diagnostics for any region. In regions with limited access to RT-PCR testing, the paper shows that using the more affordable rapid antigen test and syndromic surveillance is the next best option than only using rapid antigen testing or only syndromic surveillance.

1. Line 137: The year should be added to the months the testing was conducted.
2. There was no specification as to whether any of the cases were breakthrough infections or not. Vaccinated cases are likely to exhibit more mild symptoms and could potentially impact the symptoms chosen in the backwards selection.
3. The components selected in Model 2 (syndromic surveillance alone) were fever, diarrhea, vomit and loss of taste (excluding age). In Model 3 (rapid antigen test and syndromic surveillance), the symptoms selected were loss of taste, dry cough, wet cough and fever (including age). The symptoms and covariates included in Model 3 depend on a negative rapid antigen test. With Model 3 having two similar symptoms as Model 2, I am curious how poorly Model 3 performs without integrating the rapid antigen test result. Also, how poorly does Model 2 perform if a rapid antigen test is used? Having certain symptoms to consider with and without a rapid antigen test will be useful if access to rapid antigen tests becomes limited.
4. Line 143: There is mention of dry cough in the text, but this symptom does not appear in Table 1 (which lists cough and wet cough).
5. With the inclusion of wet and dry cough as two different symptoms, it would be interesting to see, or the authors thought on if these two symptoms were classified as a single symptom (e.g., cough). Simplifying this to a single symptom could aid in self-assessment for individuals in the community to reduce any uncertainty as to whether a cough is wet or dry (as this will impact the probability in which they would be classified as a case).
6. The coefficients for the final selected 4-symptoms models and the one-, two-, and three-symptom models are not reported in the manuscript or the supplementary material. These coefficients will provide insight into each symptom's weight in predicting whether the individual is positive for SARS CoV-2 or not.
7. The validation error for Model Class 1 (i.e., the rapid antigen-test only) likely depends most on prevalence compared to the other two models. The out-of-sample cross-entropy will depend entirely on disease prevalence in the out-of-sample cohort. The cross-entropy was determined in a temporal manner (either through the first N/k individuals or T/k days). With rising incidence over the study period, it's not surprising cross-entropy was poor. I would suggest a more random selection for calculating the cross-entropy. I do think the current approach is useful, and if there is a way also to report the prevalence of disease and corresponding cross-entropy, this analysis would complement that in Figure 3 (which has two epidemic related scenarios: rising incidence or low prevalence)
8. Model 1 considers a single rapid antigen test. The authors could analyze the impact of a secondary rapid antigen test on the results (e.g., the individual is positive if at least one of the rapid tests is positive). This second test will drastically increase the true positive rate, with only a moderate increase in the false-positive rate. Alternatively, the authors can discuss the impact of additional rapid antigen tests in diagnosis.
9. The rapid antigen test's percent positive agreement and percent negative agreement with RT-PCR is not reported in the text; these values are only reported in Figure 2.
10. Figure 2. The crosses showing the interquartile ranges make the figure very hard to read. I would suggest plotting the median with a line and using a shaded region to illustrate the uncertainty in the estimates.
11. Figure 2. The legend should indicate that the rapid antigen test is a dot and not a line.
12. There is no error in model class one in any of the figures. The error for Model class one will be dictated by the sample size used to compute the percent positive agreement and the percent negative agreement.

13. I expect the symptoms selected in Model 3 (rapid antigen test and syndromic surveillance) would depend on the rapid antigen test used. I suggest integrating this minor point into the paragraph containing lines 214-234.

14. The importance of each symptom is briefly described in text within the manuscript. To increase the clarity of Figures 1-3, I would suggest summarizing the symptoms for each model in a table.

15. The suggested use of rapid antigen tests is often 5 to 7 days after symptom onset. If the duration of symptoms to the sample is available from the dataset, it could be a helpful covariate in Model 3 (rapid antigen test and syndromic surveillance).

Reviewer #2 (Remarks to the Author):

Attached a word document. Suggest major revisions.

Reviewer #3 (Remarks to the Author):

This manuscript explores the possibility of using syndromic data to inform diagnosis when used in combination with rapid antigenic testing. The paper is not always very clear and the writing could improve, both in term of style and clarity.

The idea of combining information to inform diagnosis is not new, and the authors found that increasing the amount of data increase the accuracy of the diagnosis. This is not too surprising, as increasing information should lead to improve predictions, if the variables included are somehow informative. While the paper is interesting, the methods are hard to follow, results difficult to interpret due to unclear methods. Thus judging the clinical/epidemiological relevance in term of improved predictions is still hard.

This idea of adapting the strategies for a given local scenario is more novel, but at the moment, it is unclear what the recommendations are. Perhaps better defining what is being optimised would help.

While always a bit hard when methods are at the end, the article should still aim to explaining the results and give enough information for those results to be understood. For instance, L139, talking about model classes is unhelpful and it would be much better to give a bit of background and perhaps include the model class in `()`. Also, what model is used? Logistic regression? How the best model is reached. I realise that some information is included later in the methods, but the reader need to be able to understand as presented.

Relying on model entropy for selection is fine, but it would be helpful to give other measures of fit that are easier to understand.

I feel that it would be helpful to be more explicit and to give concrete results, e.g. number of false positive/negative for well assigned for the full model. Then same numbers under different scenario of prevalence. Prevalence scenario need to be clearly outlined in the text before the methods so the reader can understand.

When talking about large scale testing, I believe the authors think of testing for clinical purpose, identifying cases, rather than identifying trend in incidence within a population. The focus on individual diagnosis should be made clear from the introduction.

The consequences of misdiagnosis could be explored more? What are the implications in term of the effective reproduction number? Ultimately, from a public health perspective, reducing transmission would be the aim in optimising a surveillance system. Of course this might be in conflict with the more clinical approach? Currently, in the manuscript it is not very clear what is optimised.

Other comments

L45: were fitted?

L106: unclear

L127: next best methods, or use of a single method? The best method would be PCR?

L136: what is measured as case positivity? Unclear, is a prevalence of infection? Based on what data? How was it measured? PCR confirmed? How test were collected? Representative sample?

Table 1: what is the percentage asymptomatic from those?

L144 to 148: unclear what is meant there.

L155-161: could be more quantitative about the results

L168: how is the quoted error measured?

Page 7: Unclear what are the scenarios?

L180: what covariates if symptoms are not used?

L183: performance, precision; how are those measured?

Figure 3: very hard to understand what is plotted. Figure should be as stand-alone as possible. This is very far from a good balance.

L201: give the percentages themselves.

L208-9: unclear what is meant here.

L233: how can prevalence be used as covariate? The whole argument being that prevalence is unknown due to lack of testing?

L245-51: rather vague statements, would be good to be very explicit.

L278: from this, the prevalence of infection within the sample is going to be extremely biased. No statement of prevalence of infection can be made, right?

Figure 4: I strongly advise against using model class #, and instead being explicit about syndromic, RAT, combined surveillance models. It would really affect word count and be more explicit.

L302: unclear, does this mean RAT positivity over-rule the decision of the model? How results change without such procedure? Sounds like a 2 steps model then, where the combined model is only applied to the negative RAT, but informed by the whole dataset?

Methods: what about priors?

L323: how coarse correlation measured? Need to be explicit, Pearson correlation? Retain variable for a give p-value threshold?

Table 2 and associated text: how the requirement fits in? how is it used? Unclear. Reporting the both errors in all situation would help. Then commenting on the best strategy.

Unclear whether the performance of those strategies are context dependent. They are evaluated using one dataset that if I understand well fits with scenario 2? If incidence was much lower, wouldn't this change the performance?

Methods: the cross validation methodology is essential to interpret the results and should be included in the methods, not just the appendix.

Where all performance indicators come from? Are they out-of-sample indicators?

Reviewer #2 (Remarks to the Author):

Reviewer comments for Combining Rapid Antigen Testing and Syndromic 2 Surveillance Improves Community-Based COVID-19 3 Detection in Low-to-Middle-Income Countries - NatComm

In this paper, Chadwich and colleagues demonstrate in an LMIC context, how imperfect but rapid diagnostic tests can be deployed at scale combined with active surveillance for better identification of a Covid positive case. Rapid-antigen-tests and PCR validation was performed on 1172 symptomatically identified individuals at home. Using PCR as gold standard, prediction models were created using rapid-antigen-test results, syndromic data, and their combination.

Overall, a pragmatic use of existing data with some limitations. I have a few major and minor comments listed below:

1. Line 49-51 in the abstract - Models combining rapid-antigen-test and syndromic data 50 yielded equal-to-better performance to rapid-antigen-test-only models across all scenarios." – can some numeric results be added in the abstract.
2. Line 52-52 – "reduces false positive and -negative diagnoses to match local requirements"- add numeric results
3. Line 131- Add % for whom complete data was available for analyses.
4. Figure 1 demonstrating cross entropy values can be moved to supplement
5. Table 1: Is wet cough different from cough, or were they mutually inclusive categories?
6. Line 278- Participants were selected for testing if they were over 15 years old, had a fever (>38°C), and one or more of 14 symptoms listed in Table 1."... if fever was a criterion, why is it present in only 89% of the participants (Table 1)? Please also add in Table 1, the number who were asymptomatic...presuming you had those. What criterion for testing was used for those? Was there any change in the CST data collected app during the study period?
7. For **Model Class 2**, Bayesian multivariate probit model was used, for Model Class 1 and 3, which models were used?
8. How were data split into training and testing dataset for all three model classes.
9. While evaluating predictive performance of models on scenarios, why **15%** value was used as threshold (p^*) in calculating error rates. Any literature backing this up?
10. Lines 210-23 : "Applying our framework to the thousands of cases confirmed daily in Dhaka by PCR, mass deployment 212 of rapid antigen tests with syndromic surveillance can catch tens to thousands 213 of cases that would otherwise be missed." – can you please give numerically, based on a hypothetical assumption of positivity rate in a pandemic scenario.
11. In discussion, a potential limitation is the generalizability of the model for changing disease epidemiology such as varying symptomatic presentation as has particularly been seen with Omicron, please highlight accordingly. Another potential limitation is not including vaccination. Some contexts about the vaccination rates and how that may impact symptomatic presentation should be given either in the introduction or the discussion.

Overall, I believe it is a valuable study that will make a significant contribution to the evidence base. I am happy to review any responses to the comments if needed.

We would like to thank all three reviewers for the useful and insightful comments. Our point-by-point responses are denoted below by the use of italics.

Reviewer #1 (Remarks to the Author):

Chadwick et al. used a machine learning approach to determine which symptoms best identify a COVID-19 case (equivalent to RT-PCR positive) with and without the aid of a rapid antigen test. In the discussion, the authors mention that the final symptoms and covariates are likely specific to the small unique sub-population examined, which is a drawback for the broad applicability of their particular results from the syndromic surveillance. However, the approach and methodology used can be applied to improve diagnostics for any region. In regions with limited access to RT-PCR testing, the paper shows that using the more affordable rapid antigen test and syndromic surveillance is the next best option than only using rapid antigen testing or only syndromic surveillance.

We would like to thank the referee for their insightful comments below and for recognising the broad applicability and importance of our validated workflow.

1. Line 137: The year should be added to the months the testing was conducted.

We have changed the original line “(19th-26th May) to the last week (4th-11th July) of the study” to “(19th-26th May 2021) to the last week (4th-11th July 2021) of the study”. New line number 182-183

2. There was no specification as to whether any of the cases were breakthrough infections or not. Vaccinated cases are likely to exhibit more mild symptoms and could potentially impact the symptoms chosen in the backwards selection.

We agree with the reviewer’s comment that vaccination status would likely affect the symptom profile. At the time of our study in Dhaka, Bangladesh, none of the patients presenting had been vaccinated. We have added the following at line 177-178 to clarify this:

“No participants had been vaccinated as the study pre-dated mass vaccination in low-income communities in Dhaka. “

A benefit of our combined model over, for example, the RAT result alone, is that it would be very easy to explicitly account for vaccination status by including it as an additional response dimension, allowing the symptom profile for vaccinated and unvaccinated patients to differ. To clarify this, we have added the following paragraph to the discussion, lines 308 to 324:

*“The modelling frameworks allow for the potential inclusion of additional covariates where they are collected reliably. These covariates may define different sub-populations in which we expect the relationships between symptoms and infectious status to differ. For example, vaccinated patients would be expected to exhibit fewer and milder symptoms than unvaccinated patients. By including vaccination status alongside symptoms within the model, the model can share information between the two groups while allowing the relationships to differ where this improves prediction. Similar approaches could be taken to incorporate rapid antigen test manufacturer, recent disease prevalence or time since symptom onset. Furthermore, using a modelling framework allows explicit estimation and exploration of these differences, rather than relying on *post hoc* analysis of misdiagnosis rates (for example,*

[24]). *When a particular data source is found to have good predictive power, it would be useful to identify whether this could target further data collection. For example, the low false-positive rate of rapid antigen tests means that, if affordable, serial-testing of the same individual could increase true positive detections without a major impact on accuracy. “*

3. The components selected in Model 2 (syndromic surveillance alone) were fever, diarrhea, vomit and loss of taste (excluding age). In Model 3 (rapid antigen test and syndromic surveillance), the symptoms selected were loss of taste, dry cough, wet cough and fever (including age). The symptoms and covariates included in Model 3 depend on a negative rapid antigen test. With Model 3 having two similar symptoms as Model 2, I am curious how poorly Model 3 performs without integrating the rapid antigen test result. Also, how poorly does Model 2 perform if a rapid antigen test is used? Having certain symptoms to consider with and without a rapid antigen test will be useful if access to rapid antigen tests becomes limited.

The relationship between test negativity, test supply, and symptoms is interesting and important. The method described by the reviewer (seeing how Model 3 would perform without the rapid antigen test result) is functionally the same as fitting Model 2, as both models are selected from the same set of symptoms. To clarify this, we have modified the discussion to read (lines 293-295):

“From the same symptom and risk factor set, different variables were retained for different model classes, despite data being collected over a short period from the same population.”

To reflect the importance of methods which are robust to issues such as test supply problems, we have also added the following to the discussion (lines 333:341):

“The algorithm of the app could therefore be modified to reflect local epidemiological requirements, informed by local case rates and the considered cost/benefits of misdiagnosis, thereby facilitating adaptation to new variants or even new diseases. Similarly, if a source of data becomes unavailable then the underlying model can be changed to reflect this. For example, if there are rapid antigen test supply problems, the app could deploy Syndromic-only which uses the same data as Syndromic-RAT Combined, without relying on the rapid antigen test and the combined model could be retrained on tests from different manufacturers with different performance characteristics.”

4. Line 143: There is mention of dry cough in the text, but this symptom does not appear in Table 1 (which lists cough and wet cough).

Thank you for drawing this typo to our attention. This has now been corrected in Table 1 (page 8).

5. With the inclusion of wet and dry cough as two different symptoms, it would be interesting to see, or the authors thought on if these two symptoms were classified as a single symptom (e.g., cough). Simplifying this to a single symptom could aid in self-assessment for individuals in the community to reduce any uncertainty as to whether a cough is wet or dry (as this will impact the probability in which they would be classified as a case).

We agree that the ability to diagnose a wet from dry cough may lead to these symptoms being less reliable separately than together. As such, we reran our whole model selection process with dry cough, wet cough and any cough. Interestingly, the model selection process removed “any cough” but kept the other two for the Syndromic-RAT Combined

model implying that even in-community reporting of the two cough types is of sufficient quality to distinguish the two as symptoms.

6. The coefficients for the final selected 4-symptoms models and the one-, two-, and three-symptom models are not reported in the manuscript or the supplementary material. These coefficients will provide insight into each symptom's weight in predicting whether the individual is positive for SARS CoV-2 or not.

We agree that it is useful to know the contribution of different symptoms, particularly if changes in these quantities can be observed over time. To demonstrate this, we have added the following to "Supplementary Materials (Additional Results: Correlation Estimates)":

"The relationship between symptoms and results can only be understood through the full correlation matrix. Here we present the median correlations for the four symptom SyndOnly and SyndRAT models. These results should not be used to prioritise future data collection because the most predictive symptoms are liable to change with time (e.g., the emergence of new COVID variants) and context (e.g., broader vaccination levels)."

However, we do strongly advise against over-interpreting or determining policy using these coefficients as they will be strongly dependent on the focal population and the time of observation.

7. The validation error for Model Class 1 (i.e., the rapid antigen-test only) likely depends most on prevalence compared to the other two models. The out-of-sample cross-entropy will depend entirely on disease prevalence in the out-of-sample cohort. The cross-entropy was determined in a temporal manner (either through the first N/k individuals or T/k days). With rising incidence over the study period, it's not surprising cross-entropy was poor. I would suggest a more random selection for calculating the cross-entropy. I do think the current approach is useful, and if there is a way also to report the prevalence of disease and corresponding cross-entropy, this analysis would complement that in Figure 3 (which has two epidemic related scenarios: rising incidence or low prevalence)

We thank the reviewer for drawing attention to the role of prevalence in out-of-sample prediction, however, we believe that it is important to present method performance conservatively. Using random selection of samples for cross-validation would overstate the power of the modelling approaches as it would ignore the changing nature of the disease and its management.. One of the challenges of all modelling approaches is that we need to predict into the future based on past data but the nature and management of the disease are changing over time. Temporal cross-validation is the only way to assess applied model performance, because the real-world prediction problem is temporal in nature. To emphasise this point in the text, in the Methods: Model Selection section we have added the following text at lines 436-440:

"We structured the cross-validation temporally to reflect the real-world prediction problem: using recent testing data to predict new cases. Due to the changing nature of the disease and its management over time, using unstructured cross-validation would result in an overstatement of model performance."

8. Model 1 considers a single rapid antigen test. The authors could analyze the impact of a secondary rapid antigen test on the results (e.g., the individual is positive if at least one of the rapid tests is positive). This second test will drastically increase the true positive rate,

with only a moderate increase in the false-positive rate. Alternatively, the authors can discuss the impact of additional rapid antigen tests in diagnosis.

We agree that multiple rapid antigen tests, if affordable, bring far more benefit than cost. To reflect this, we have added the following to the discussion at lines 319-324:

“When a particular data source is found to have good predictive power, it would be useful to identify why this is the case and whether this could target further data collection. For example, the low false-positive rate of rapid antigen tests means that, if affordable, multiple-testing of the same individual could increase true positive detections without a major impact on accuracy. “

9. The rapid antigen test's percent positive agreement and percent negative agreement with RT-PCR is not reported in the text; these values are only reported in Figure 2.

These values have now been added to the Results section at lines 216-219 as follows:

“The RAT-only model is a binary test (rapid-antigen-test positive or negative) and so the ROC is a single value, not a curve, with false positive rate of 0.02 and a false negative rate of 0.45.”

10. Figure 2. The crosses showing the interquartile ranges make the figure very hard to read. I would suggest plotting the median with a line and using a shaded region to illustrate the uncertainty in the estimates.

We agree that the plot looks challenging as presented, however, we feel a smooth curve would misrepresent the model ROCs as the values are not in fact continuous. To emphasise this, we have overlaid dark points at the median values to make them more visible and have added the following to the figure caption:

“While ROC plots are often plotted as curves, we do not have continuous probability values due the binary nature of predictor symptoms. This is important as discontinuity in the probabilities impacts the sensitivity of the model to classification thresholds, such as those used in the scenarios below.”

11. Figure 2. The legend should indicate that the rapid antigen test is a dot and not a line.

The Figure 2 legend now states:

“In the RAT-only model the ROC is a single value (i.e. a dot rather than a line) as the binary test has a single sensitivity and specificity.”

12. There is no error in model class one in any of the figures. The error for Model class one will be dictated by the sample size used to compute the percent positive agreement and the percent negative agreement.

Model Class 1 is not a statistical model, the misclassification rates are deterministic properties of the data, rather than estimates, and so do not have a meaningful error.

13. I expect the symptoms selected in Model 3 (rapid antigen test and syndromic surveillance) would depend on the rapid antigen test used. I suggest integrating this minor point into the paragraph containing lines 214-234.

We have integrated this point into the paragraph added in response to your point 2, specifically at line 315-317:

“Similar approaches could be taken to incorporate rapid antigen test manufacturer, recent disease prevalence or time since symptom onset.”

14. The importance of each symptom is briefly described in text within the manuscript. To increase the clarity of Figures 1-3, I would suggest summarizing the symptoms for each model in a table.

See response to point 6.

15. The suggested use of rapid antigen tests is often 5 to 7 days after symptom onset. If the duration of symptoms to the sample is available from the dataset, it could be a helpful covariate in Model 3 (rapid antigen test and syndromic surveillance).

Unfortunately we do not have these data, but have modified the text per the response to point 13.

We would like to thank the reviewer for their valuable feedback.

Reviewer #2 (Remarks to the Author):

Reviewer comments for Combining Rapid Antigen Testing and Syndromic 2 Surveillance Improves Community-Based COVID-19 3 Detection in Low-to-Middle-Income Countries - NatComm

In this paper, Chadwich and colleagues demonstrate in an LMIC context, how imperfect but rapid diagnostic tests can be deployed at scale combined with active surveillance for better identification of a Covid positive case. Rapid-antigen-tests and PCR validation was performed on 1172 symptomatically identified individuals at home. Using PCR as gold standard, prediction models were created using rapid-antigen-test results, syndromic data, and their combination.

Overall, a pragmatic use of existing data with some limitations. I have a few major and minor comments listed below:

Thank you to the reviewer for their useful comments and their excellent summary of the paper.

1. Line 49-51 in the abstract - Models combining rapid-antigen-test and syndromic data 50 yielded equal-to-better performance to rapid-antigen-test-only models across all scenarios.” – can some numeric results be added in the abstract.
2. Line 52-52 – “reduces false positive and -negative diagnoses to match local requirements”- add numeric results

Response to Points 1 and 2: We agree that the abstract would be much improved by the addition of numbers, however, we are wary that to summarise all model-scenario combinations would overload the abstract. To balance these two demands, we have now highlighted what we feel is the key example (corresponding to pandemic growth, the scenario in which these techniques are both most useful and most likely to be needed) (lines 65-70):

“The best performance was in the scenario corresponding to rapid epidemic growth, in which the combined data model has a false negative rate 22 (IQR: 19-25) percentage points lower than the rapid antigen test model. Although the syndromic model matches the combined model’s false negative rate, its false positive rate is 33 (IQR: 30-33) percentage points higher.”

3. Line 131- Add % for whom complete data was available for analyses.

Line 171 now reads: Of 1241 participants enrolled by community support teams across Dhaka, 1172 (94%) had complete data available.

4. Figure 1 demonstrating cross entropy values can be moved to supplement

We agree that Figure 1 does not currently convey its message clearly enough for the main text of the article. However, we believe that it is important to demonstrate model comparison at the scale of probability, generalised classification (ROC curves) and scenario-specific performance. To better illustrate this point, we have modified the caption of Figure 1 (page 10) to include:

“Predictive performance of candidate models measured using out-of-sample cross-entropy. Cross-entropy shows the most generalised-level of model predictive power, assessing performance in the probability scale without requiring decisions to be made regarding classification thresholds.”

5. Table 1: Is wet cough different from cough, or were they mutually inclusive categories?

Thank you for drawing our attention to this error. Here “cough” was intended to refer to dry cough, we have modified Table 1 (page 8) to reflect this.

6. Line 278- Participants were selected for testing if they were over 15 years old, had a fever (>38°C), and one or more of 14 symptoms listed in Table 1.”... if fever was a criterion, why is it present in only 89% of the participants (Table 1)?

Thank you for drawing our attention to the ambiguity here. There are two types of fever that were used. In the symptom list, this refers to self-reported, ongoing fever. In the selection criteria, it is fever at the point of assessment. To remove these ambiguities we have revised Table 1 to list “Ongoing Fever” as the symptom and now refer to it as such when discussed in the text, and at line 391-392, the selection criteria now reads:

“Participants were selected for testing if they were over 15 years old, had a fever (>38°C) at the point of assessment, and one or more of 14 symptoms listed in Table 1,”.

Please also add in Table 1, the number who were asymptomatic...presuming you had those. What criterion for testing was used for those?

No asymptomatic patients were included in this study as resources were limited and the government was prioritising symptomatic patients (particularly in these low-income settings with limited access to care). To better clarify this point, in the introduction paragraph addressing asymptomatic patients we have added the following line (113-115):

“Reliable diagnosis of symptomatic cases of COVID-19, therefore, is a priority in many settings and is the focus of this paper.”

We have also added the following statement to the results section at line 178-179:

“only symptomatic patients were included in this study because they were the local government priority for support”

Was there any change in the CST data collected app during the study period?

The app was updated prior to the study and there were no updates during the study period.

7. For **Model Class 2**, Bayesian multivariate probit model was used, for Model Class 1 and 3, which models were used?

Thank you for drawing our attention to the lack of clarity here. We have modified the “Model Structure” section of the methods as follows (lines 306-432, bold is used to highlight changes here and is not present in the text):

“We developed three model classes using: 1. the rapid-antigen-test result; 2. the syndromic data, and 3. the two data sources combined (Figure 4). We identified cases by PCR. **As RAT-only used the rapid-antigen-test result, no statistical model is needed.** For Syndromic-only, we used a Bayesian multivariate probit model,[31] with multivariate referring to multiple response variables. The multivariate probit structure allows the model to account for the binary and correlated nature of the symptoms, while conditioning on the risk factors of age and gender, thereby improving over models which implicitly assume independence between symptoms. By using a Bayesian formulation, we generate full posteriors for our parameter estimates, allowing natural quantification of uncertainty. We chose minimally informative priors, with standard normals for the covariates and intercepts and a flat LKJ distribution for the correlation matrix (described in more detail in Supplementary Materials: Statistical Methodology).

For Syndromic-RAT Combined, we use a hurdled multivariate probit.

The approach exploits the specificity of rapid antigen tests by treating rapid test-positives as cases. While this sounds like a strong assumption, this simply translates in practice to telling all rapid test-positive individuals to assume they have COVID-19. Rapid-antigen-test-negative individuals are then modelled using the sensitive syndromic approach of Syndromic-only to capture PCR-positives missed by the rapid antigen test. This approach leverages the potentially different syndromic profiles of PCR-positive patients who are rapid-antigen-test-positive and -negative, allowing the model to adapt solely to the latter. The models were fitted to the data using Bayesian inference techniques based on Hamiltonian Monte Carlo in the Stan programming language. Further technical details and model equations are presented in Supplementary Materials: Statistical Methodology.

8. How were data split into training and testing dataset for all three model classes.

We have added the following paragraph to improve our explanation of how data were split (lines 434-440):

“For model selection and all measures of performance, we used out-of-sample, temporal cross-validation, where training and testing data are separated based on time. We structured the cross-validation temporally to reflect the real-world prediction problem: using recent testing data to predict new cases.”

9. While evaluating predictive performance of models on scenarios, why 15% value was used as threshold (p^*) in calculating error rates. Any literature backing this up?

The 15% threshold is used as an example. To clarify this we have modified the text to read (line 734-737):

“if the requirement were, hypothetically, that an error rate should be a maximum 15%, the threshold that produces an error rate below 15% but as close to 15% as possible will be chosen”

The 20% thresholds which were used in our analysis were determined in discussion with government agencies. To better emphasise this, we have modified the caption for Table 2 (page 19) to read:

“These requirements were determined through discussion with colleagues at IEDCR using internal resource projections”

10. Lines 210-23 : “Applying our framework to the thousands of cases confirmed daily in Dhaka by PCR, mass deployment 212 of rapid antigen tests with syndromic surveillance can catch tens to thousands 213 of cases that would otherwise be missed.” – can you please give numerically, based on a hypothetical assumption of positivity rate in a pandemic scenario.

To more clearly make this point , we have given an estimate of the number of cases that would be missed per 100 tests deployed and a 20% case positivity rate as follows (lines 274-279):

“In real terms, at the end of our study, there was a 20% case positivity rate in Bangladesh. By applying our framework under the "Epidemic Growth" scenario, for every 100 rapid antigen tests, our approach would capture an additional 7 cases. In a country deploying millions of tests per week, the result is catching tens of thousands of cases that would otherwise be missed.”

11. In discussion, a potential limitation is the generalizability of the model for changing disease epidemiology such as varying symptomatic presentation as has particularly been seen with Omicron, please highlight accordingly. Another potential limitation is not including vaccination. Some contexts about the vaccination rates and how that may impact symptomatic presentation should be given either in the introduction or the discussion.

Please see response to Reviewer 1, point 2 above.

Overall, I believe it is a valuable study that will make a significant contribution to the evidence base. I am happy to review any responses to the comments if needed.

We would like to thank the reviewer for their time and valuable feedback.

Reviewer #3 (Remarks to the Author):

This manuscript explores the possibility of using syndromic data to inform diagnosis when used in combination with rapid antigenic testing. The paper is not always very clear and the writing could improve, both in term of style and clarity.

The idea of combining information to inform diagnosis is not new, and the authors found that increasing the amount of data increase the accuracy of the diagnosis. This is not too surprising, as increasing information should lead to improve predictions, if the variables included are somehow informative. While the paper is interesting, the methods are hard to follow, results difficult to interpret due to unclear methods. Thus judging the clinical/epidemiological relevance in term of improved predictions is still hard.

This idea of adapting the strategies for a given local scenario is more novel, but at the moment, it is unclear what the recommendations are. Perhaps better defining what is being optimised would help.

Although the idea of corroboration from different sources of information is not new, we firmly believe that the formal integration of different diagnostic information presented here, is genuinely novel and generalisable. Admittedly, this more complex statistical methodology requires more effort to explain. We are grateful to the referee for pointing out areas in the MS where this will be most helpful. We have improved the flow of the paper per the methods-last format of the journal, and expanded the discussion on the more novel aspects of the paper.

One aspect we were keen to highlight is that all diagnostics make decisions about how to balance error types, but ours makes this explicit and interrogable. We have tried to better describe this and highlight the challenges in the Introduction (Lines 128-147):

“The typical approach is to balance sensitivity and specificity to maximise the number of correct classifications and assume that both misclassification types are equally costly. The costs of false positives and false negatives, however, vary enormously depending on the intersection of perspective, economic and epidemiological concerns. An individual may be motivated to secure a false negative diagnosis if there is insufficient support for self-isolation. In contrast, at the government level, false negatives may be acceptable if the economic cost of supporting those individuals is less than the cost of accelerating case rates. The epidemiological context will also alter the impact of false positives and false negatives. For example, if the disease is prevalent or increasing the priority of both individuals and governments may be to curb transmission and reduce impacts as quickly as possible. In this instance, false negatives have an outsized and costly impact by increasing the number of contact events occurring in the population and delaying control measures by underestimating epidemic size.[19] In contrast, under low prevalence, false negatives will be correspondingly low so even a high false negative rate (low sensitivity) will have modest impact, but small decreases in specificity will lead to a large number of expensive false positives.[20] In practice the situation will be more nuanced and modulated by testing capacity constraints, requiring a balance to be struck.[17]”

and Discussion as follows (lines 342-364):

“One of the key innovations of this framework is the ability to adapt the diagnostic to local populations and their needs. To achieve this, we need good quality, local data collection and to understand the costs of sensitivity and specificity. The costs of false negatives and false positives vary greatly depending on epidemic context, and balancing the treatment of

individuals with control of the health burden at a societal level.[27] Similarly, the market price of interventions can fluctuate depending on demand, aid funding and global trends.[28] In practice, the costs of rapid antigen tests are likely to be up to an order of magnitude lower than PCR when considering the additional infrastructure and personnel. Access to testing (RAT or PCR) needs to be considered as part of weighing up the costs and benefits of surveillance approaches.[29] Understanding how to measure and balance these demands requires insights from economists, epidemiologists, social scientists and policy-makers, and is an area of active research. [30] Given the degree of complexity, it is tempting to rely on methods that do not openly require a decision to be made about the relative costs of the different misclassification types. However, rather than removing the complex cost structures involved, such methods simply hide them. All methods place a balance on false positives and negatives implicitly, our hope is that by requiring these decisions to be made explicitly, they are more readily challenged, researched and improved upon. Similarly, the need for local data collection should not be seen as a weakness of the method, but rather a welcome requirement that allows us to directly assess intervention success and biases.”

While always a bit hard when methods are at the end, the article should still aim to explaining the results and give enough information for those results to be understood. For instance, L139, talking about model classes is unhelpful and it would be much better to give a bit of background and perhaps include the model class in ‘()’. Also, what model is used? Logistic regression? How the best model is reached. I realise that some information is included later in the methods, but the reader need to be able to understand as presented.

We agree that the methods late in the text make it challenging to parse the different models early on. We have modified the results section (where the different models are first introduced) as follows to include a brief summary of the model type (multivariate probit and hurdled multivariate probit), the selection criteria and the relevant section of the methods (lines 185-190):

“Backwards model selection using strength of posterior correlation with outcome (Methods (Statistical Modelling: Model Selection)) for both the multivariate probit syndromic data only model and the thresholded multivariate probit syndromic data with rapid antigen test result (hereafter the Syndromic-only and Syndromic-RAT combined) models showed a marked decline in predictive power at more than 4 symptoms.”

We also emphasise that model performance was compared using temporally structured cross-validation by introducing this alongside the type of performance metric (lines 202-203): “In the comparison of predictive performance under out-of-sample temporal cross-validation”

Relying on model entropy for selection is fine, but it would be helpful to give other measures of fit that are easier to understand.

We agree that model entropy was insufficiently explained, particularly in Figure 1, so have added a more detailed explanation in the caption and included a null comparison (for a random classifier). We are unaware of easier to understand measures of predictive performance that work in the probability scale and that are also (statistically) proper scoring rules. The updated caption now reads (page 10):

“Predictive performance of candidate models were measured using out-of-sample cross-entropy. Cross-entropy shows the most generalised-level of model predictive power, assessing performance in the probability scale without requiring decisions to be made

regarding classification thresholds. A cross-entropy of zero indicates a model that predicts with certainty the correct result each time. A random classifier for the problem scored 11.54.”

I feel that it would be helpful to be more explicit and to give concrete results, e.g. number of false positive/ negative for well assigned for the full model. Then same numbers under different scenario of prevalence. Prevalence scenario need to be clearly outlined in the text before the methods so the reader can understand.

Thank you for this suggestion for how to better communicate model performance in an intuitive way. We have added an additional table (Table 2) to the results section which presents false positive and negative numbers for each model-scenario combination under a range of case positivity rates. We draw attention to this result in the main text as follows (lines 244-247):

“The results for each scenario-model combination can be translated into numbers of misclassifications per 1000 tests if the case positivity rate is known. We present this in Table 3 (Supplementary Materials: Additional Results) for low- (5%), average-(20%) and high- (35%) test positivity rates in Bangladesh.”

And in the Supplementary Materials: Additional Results:

“False positive and false negative rates can only be translated into numbers of people affected if the case positivity rate is known. To demonstrate how the numbers of misclassifications change for the same false positive and false negative rates, we have scaled these numbers with respect to low- (5%), average- (20%) and high- (35%) test positivity rates in Bangladesh in Table 3”

When talking about large scale testing, I believe the authors think of testing for clinical purpose, identifying cases, rather than identifying trend in incidence within a population. The focus on individual diagnosis should be made clear from the introduction.

We concur that methods for diagnosing individuals (e.g. chest CT, PCR scans) and for monitoring population levels of the disease (e.g. wastewater sampling, hospital admissions monitoring) do not always overlap. However, governments in low- and middle-income countries (and indeed in HICs) are frequently trying to achieve both with the same tests. To draw greater attention to this, we have modified the first paragraph of the introduction to read (lines 90-93):

“Improving COVID-19 diagnosis is a priority and, therefore, requires us to better harness imperfect but fast and inexpensive methods, particularly for individual diagnosis but also for population-level surveillance.”

The consequences of misdiagnosis could be explored more? What are the implications in term of the effective reproduction number? Ultimately, from a public health perspective, reducing transmission would be the aim in optimising a surveillance system. Of course this might be in conflict with the more clinical approach? Currently, in the manuscript it is not very clear what is optimised.

We agree that different approaches may be needed to balance the rights and needs of the individual with the public health demands of the society (a parallel of monitoring at the society level), however, we believe it is the role of local policy makers to determine this balance. For that reason, our framework does not, a priori, favour one over the other.

Rather, quantities such as the effective reproduction number could (and should) be used in assessing the cost of false positives or negatives. To emphasise this important tradeoff, we have added the paragraphs (lines 128-147 and 342-364) highlighted earlier in our response to your comments.

Other comments

L45: were fitted?

Corrected

L106: unclear

Original line: "The typical approach is to maximise the number of correct classifications and assume that both misclassification types are equally costly."

Revised line: "The typical approach is to balance sensitivity and specificity to maximise the number of correct classifications and assume that both misclassification types are equally costly." (Lines 128-230)

L127: next best methods, or use of a single method? The best method would be PCR?

PCR is the most accurate method but arguably not the best due to its expense. To be more clear we have revised the the original line as follows

Original line: "Overall, we show that the optimised combined models achieve equal-to-much-lower error rates than the next best method in all metrics"

Revised line: "Overall, we show that the optimised combined models achieve equal-to-much-lower error rates than the rapid antigen test- or syndromic surveillance-only in all metrics" (lines 164-166)

L136: what is measured as case positivity? Unclear, is a prevalence of infection? Based on what data? How was it measured? PCR confirmed? How test were collected? Representative sample?

Thank you for drawing our attention to this, we have clarified that this was measured by PCR and added the relevant data citation [180-183]:

"Case positivity measured by PCR in Dhaka increased from 15.8% to 23.8% from the first (19th-26th May 2021) to the last week (4th-11th July 2021) of the study, corresponding to prevalence rising from 1.4 to 13.8 confirmed cases per 100 000 people [21]."

Table 1: what is the percentage asymptomatic from those?

0%. No asymptomatic patients were included in this study as resources are so limited in low-to-middle income countries that symptomatic patients have to be prioritised. To better clarify this point, in the introduction paragraph addressing asymptomatic patients we have added the following line (lines 113-115):

"Reliable diagnosis of symptomatic cases of COVID-19, therefore, is a priority in many settings and is the focus of this paper."

We have also added the following statement to the results section (lines 178-180):

“only symptomatic patients were included in this study as they were the local government priority for support”

L144 to 148: unclear what is meant there.

Original line: “The symptoms are listed in the order they removed through model selection (i.e. all four symptoms were retained in the four symptom model, the first was removed in the three symptom model, the second was also removed in the two symptom model etc.).

Revised line: “The symptoms are listed in reverse order of importance as determined by model selection (i.e. all four symptoms were retained in the four symptom model, the first was removed in the three symptom model, the second was also removed in the two symptom model etc.)” (lines 193-196)

L155-161: could be more quantitative about the results

We are wary of attempting to summarise ROC curves quantitatively as there are no sufficient statistics, with many metrics such as area under the curve being both misleading, and relying on an assumption of continuous probability which is violated in this model.

L168: how is the quoted error measured?

We have now cited the relevant table.

Page 7: Unclear what are the scenarios?

Thank you for drawing our attention to this lack of clarity. We have now provided brief summaries of each scenario when they are introduced and signposted the scenarios definition table for more details, e.g. lines 226-227:

“In Scenario 1 (“Agnostic”, wherein the correct classification is maximised, assuming equal costing of false positives and false negatives, see Table 2)”

L180: what covariates if symptoms are not used?

We have revised this to say (lines 250-251):

“with no symptoms but an intercept and age and gender as covariates”

L183: performance, precision; how are those measured?

We have clarified that this is in term of scenario defined error, as follows (lines 251-255):

“For the models that used syndromic data across all scenarios, the number of symptoms made relatively little difference within the final four candidate models in terms of median performance (with respect to error, see Figure 3 and Table 2)”

Figure 3: very hard to understand what is plotted. Figure should be as stand-alone as possible. This is very far from a good balance.

We agree that our previous figure caption provided insufficient explanation. We have revised the caption as follows (page 12):

“Performance of models under three epidemiological scenarios. In the Agnostic Scenario, the model is optimised to maximise the correct classification rate with error measured as the sum of the false positive and false negative rates. In the Epidemic Growth Scenario, a maximum false negative rate of 20% is permitted, and the error is measured as the false positive rate. In the Declining Incidence scenario, a maximum false positive rate of 20% is permitted, and the error is measured as the false negative rate. These requirements were determined through discussion with colleagues at IEDCR. The plot shows the posterior median and interquartile range for scenario-specific errors. Lower errors correspond to better model performance. There is no error rate defined for rapid-antigen-testing-only model (RAT-only) in the Epidemic Growth Scenario as the model failed to meet the requirement for that scenario (indicated by grey bar). Model classes are colour-coded, the RAT-only model is purple, Syndromic-only model is teal, and the Syndromic-RAT Combined model is yellow.”

L201: give the percentages themselves.

The percentages have now been added.

L208-9: unclear what is meant here.

Original line: “The more complex model classes are flexible, so can be tailored to specific needs, and benefit synergistically from combining rapid antigen testing with the non-specific syndromic data.”

New line: “The more complex model classes achieve this top performance across all scenarios and metrics measured here thanks to their flexibility (allowing them to be readily adapted to new situations) and their synergistic use of the higher specificity rapid antigen testing and the more sensitive syndromic data.” (lines 283-287)

L233: how can prevalence be used as covariate? The whole argument being that prevalence is unknown due to lack of testing?

While prevalence is unknown at present due to lack of testing, there are numerous situations under which prevalence data may be available for example through modelling, alternative testing, or through this testing at a short delay. As prevalence is temporally autocorrelated, recent data can still be extremely informative, and approximate data may still be predictive. This is particularly relevant if the focus is on individual diagnosis, as discussed previously in your review.

L245-51: rather vague statements, would be good to be very explicit.

Please see the revised paragraphs (Lines 128-147, and 342-364) discussed earlier in our response to your first point.

L278: from this, the prevalence of infection within the sample is going to be extremely biased. No statement of prevalence of infection can be made, right?

Yes, that's correct - this bias is a practical necessity for governments who need to target their limited resources. Alternative sampling regimes could be used with our modelling to give informed estimates of prevalence.

Figure 4: I strongly advise against using model class #, and instead being explicit about syndromic, RAT, combined surveillance models. It would really affect word count and be more explicit.

Thank you for this comment, adapting the labelling as suggested (see above) has made the paper much more readable.

L302: unclear, does this mean RAT positivity over-rule the decision of the model? How results change without such procedure? Sounds like a 2 steps model then, where the combined model is only applied to the negative RAT, but informed by the whole dataset?

This is exactly correct, thank you for pointing out this was unclear. To emphasise this point, we have modified the description as follows (lines 420-426):

“For Syndromic-RAT Combined, we use a hurdled multivariate probit. The approach exploits the specificity of rapid antigen tests by treating rapid test-positives as cases. While this sounds like a strong assumption, this simply translates in practice to telling all rapid test-positive individuals to assume they have COVID-19. Rapid-antigen-test-negative individuals are then modelled using the sensitive syndromic approach of Syndromic-only to capture PCR-positives missed by the rapid antigen test.”

Methods: what about priors?

We agree that the choice of priors is very important. We initially only described these in the supplementary materials but have now added the following to the main methods section (lines 416-419):

“We chose minimally informative priors, with standard normals for the covariates and intercepts and a flat LKJ distribution for the correlation matrix (described in more detail in Supplementary Materials: Statistical Methodology).”

L323: how coarse correlation measured? Need to be explicit, Pearson correlation? Retain variable for a give p-value threshold?

We agree this is a potential source of confusion. As this is a Bayesian analysis the full posterior correlation (i.e. normalised covariance) matrix is sampled using Hamiltonian Monte Carlo, rather than the use of a numerical point estimator (i.e. a correlation coefficient).

To better emphasise the procedure, we have added the following explanation to the modelling flowchart caption (page 20):

“A subset of symptoms are identified using the strength of posterior correlation between each symptom and PCR-status identified by the corresponding model, with the weakest correlated symptoms removed during each round of selection”

And the following to the Methods: Model Selection text (lines 448-455):

“The Bayesian multivariate probit structure common to these models directly estimates the full posterior correlation matrix for the PCR-status and other symptoms. By first using the strength of the correlation with the PCR-status (coarse selection, Figure 5) and general predictive power (fine selection, Figure 5) to narrow down the number of candidate models, and then testing those models under the epidemiological scenarios, we are more likely to choose models that generalise well to new data (see Supplementary Materials: Statistical Methodology).”

Table 2 and associated text: how the requirement fits in? how is it used? Unclear. Reporting the both errors in all situation would help. Then commenting on the best strategy.

Thank you for highlighting the lack of clarity here. We have added the following to the Table 2 caption (page 19) to better explain the role of the requirement:

“The requirement refers to a base level of performance the model must achieve, allowing the more flexible models to be adapted to meet that requirement as closely as possible (e.g. by determining a classification threshold).”

We agree that reporting all false positive/false negative rates can improve the scenario-free comparison of the models. The purpose of scenarios is to make decision-making clearer (and tradeoffs explicit) for policy-makers. For this reason, we define a requirement (usually one of the misclassification rates) and reduce the model comparison to a single error rate (usually of the other misclassification rates). If the requirement is met, we found it was not informative to policy-makers to see how well that requirement is met. We have included information on both error rates in the ROC figures and they can be seen for the scenarios in the additional results table that we added in response to your earlier point.

Unclear whether the performance of those strategies are context dependent. They are evaluated using one dataset that if I understand well fits with scenario 2? If incidence was much lower, wouldn't this change the performance?

Thank you for your comment highlighting a key point of confusion in our text. Our use of “Low Incidence” as one of the scenario names makes this more confusing so have renamed this scenario as “Declining Incidence”. In fact, although the absolute costs of errors may be linked to incidence, the relative costs are not (and this is the feature captured in our error rates). We have outlined more clear examples of this in the new introduction paragraph discussed above (lines 128-147)

Methods: the cross validation methodology is essential to interpret the results and should be included in the methods, not just the appendix.

Where all performance indicators come from? Are they out-of-sample indicators?

We agree that understanding the cross-validation methodology and performance indicators is essential. To make the methods behind both clearer, we have added the following explanation to the main text (lines 434-440):

“For model selection and all measures of performance, we used out-of-sample, temporal cross-validation, where training and testing data are separated by time. We structured the cross-validation temporally to reflect the real-world prediction problem: using recent testing data to predict new cases. Due to the changing nature of the disease and its management

over time, using unstructured cross-validation would result in an overstatement of model performance.”

We would like to thank the reviewer for their valuable feedback.

REVIEWERS' COMMENTS

Reviewer #1 (Remarks to the Author):

1. Some of the figure references in the supplement have "??"
2. I just wanted to clarify one of my comments from my initial review

The components selected in Model 2 (syndromic surveillance alone) were fever, diarrhea, vomit and loss of taste (excluding age). In Model 3 (rapid antigen test and syndromic surveillance), the symptoms selected were loss of taste, dry cough, wet cough and fever (including age). The symptoms and covariates included in Model 3 depend on a negative rapid antigen test. With Model 3 having two similar symptoms as Model 2, I am curious how poorly Model 3 performs without integrating the rapid antigen test result. Also, how poorly does Model 2 perform if a rapid antigen test is used? Having certain symptoms to consider with and without a rapid antigen test will be useful if access to rapid antigen tests becomes limited.

I was curious how well the covariates loss of taste, dry cough, wet cough and fever (including age) [i.e., the covariates selected in Model 3] perform in the syndromic surveillance alone model relative to the performance the best model selected for Model 2 (syndromic surveillance alone). Similarly, how well the covariates fever, diarrhea, vomit and loss of taste (excluding age) perform in the rapid antigen test and syndromic surveillance model relative to the best model selected for Model 3 (rapid antigen test and syndromic surveillance). This comparison will provide some indication of the relative importance of the different covariates and the resulting impact on performance under these different conditions. This analysis is not critical to the findings or methodology of the manuscript, and I leave it to the authors discretion if this analysis is something they want to pursue.

3. In the added text "Here we present the median correlations for the four symptom Syndromic-only and Syndromic-RAT Combined models." Figure 1 and Figure 2 in the supplement should be referenced.
4. Line 279. There are is a closing quotation mark at the end of the sentence after missed with no open quotation mark.

Reviewer #3 (Remarks to the Author):

I have now reviewed the manuscript and I am now happy with the modifications. I would like to commend the authors for their thorough revision and new insight into the context-dependent interpretations of their results. I believe the methods are also much more clear, which will benefit the readership of this article.

As a last comment, could I suggest that, in future, the authors submit a version of the modified manuscript with clear track changes? It makes a really big differences in spotting the changes when reading and make the revision process much easier. I understand that using Latex, this cannot be achieved as easily as when using word, but having said that, unless the manuscript has a lot of equations, I'm not entirely sure about the overall benefit of Latex (an argument that I can see I will lose in the long term...).

We would like to thank both reviewers for their insightful comments in both rounds of review. We have addressed their final comments point by point below.

Reviewer #1 (Remarks to the Author):

Thank you for your positive response to our revisions and for your original comments. We believe that incorporating these comments much improved the paper.

1. Some of the figure references in the supplement have “??”

Thank you for drawing this to our attention. We have double-checked the resubmission to make sure this mistake has not persisted.

2. I just wanted to clarify one of my comments from my initial review

“The components selected in Model 2 (syndromic surveillance alone) were fever, diarrhea, vomit and loss of taste (excluding age). In Model 3 (rapid antigen test and syndromic surveillance), the symptoms selected were loss of taste, dry cough, wet cough and fever (including age). The symptoms and covariates included in Model 3 depend on a negative rapid antigen test. With Model 3 having two similar symptoms as Model 2, I am curious how poorly Model 3 performs without integrating the rapid antigen test result. Also, how poorly does Model 2 perform if a rapid antigen test is used? Having certain symptoms to consider with and without a rapid antigen test will be useful if access to rapid antigen tests becomes limited.”

I was curious how well the covariates loss of taste, dry cough, wet cough and fever (including age) [i.e., the covariates selected in Model 3] perform in the syndromic surveillance alone model relative to the performance the best model selected for Model 2 (syndromic surveillance alone). Similarly, how well the covariates fever, diarrhea, vomit and loss of taste (excluding age) perform in the rapid antigen test and syndromic surveillance model relative to the best model selected for Model 3 (rapid antigen test and syndromic surveillance). This comparison will provide some indication of the relative importance of the different covariates and the resulting impact on performance under these different conditions. This analysis is not critical to the findings or methodology of the manuscript, and I leave it to the authors discretion if this analysis is something they want to pursue.

Thank you for clarifying your earlier point. We believe that the covariates selected are very important to the focal population as the comparison of models in the existing schemes demonstrates that models of the same complexity (i.e. number of symptoms) can have very different performance.

3. In the added text “Here we present the median correlations for the four symptom Syndromic-only and Syndromic-RAT Combined models.” Figure 1 and Figure 2 in the supplement should be referenced.

Thank you for highlighting this point, we have added this.

4. Line 279. There are is a closing quotation mark at the end of the sentence after missed with no open quotation mark.

We have now removed this, thank you for highlighting.

Reviewer #3 (Remarks to the Author):

I have now reviewed the manuscript and I am now happy with the modifications. I would like to commend the authors for their thorough revision and new insight into the context-dependent interpretations of their results. I believe the methods are also much more clear, which will benefit the readership of this article.

Thank you for your positive response to our revisions and for your original comments. We believe that incorporating these comments much improved the paper, in particular, on the key issue of understanding the model descriptions.

As a last comment, could I suggest that, in future, the authors submit a version of the modified manuscript with clear track changes? It makes a really big differences in spotting the changes when reading and make the revision process much easier. I understand that using Latex, this cannot be achieved as easily as when using word, but having said that, unless the manuscript has a lot of equations, I'm not entirely sure about the overall benefit of Latex (an argument that I can see I will lose in the long term...).

We agree that this would be a useful document both for the reviewer and author. We have found this solution for providing the document advised:

<https://tex.stackexchange.com/questions/584050/how-to-yellow-highlight-differences-of-two-tex-documents-automatically-in-latex> (in case you wish to share with future LaTeX users).